# Impact of increased kidney function on clinical and biological outcomes in real-world patients treated with Direct Oral Anticoagulants

**Mariana Corrochano**[1,2]*, **René Acosta-Isaac**[1,2], **Melania Plaza**[2], **Rodrigo Muñoz**[2], **Sergi Mojal**[2], **Carla Moret**[1,2], **Joan Carles Souto**[1,2]

**1** Thrombosis and Haemostasis Unit, Hospital de la Santa Creu i Sant Pau, Barcelona, Spain, **2** Intitut d'Investigacions Biomèdiques Sant Pau (IIB-Sant Pau), Barcelona, Spain

* marianacorrochanof@gmail.com

**Data Availability Statement:** All relevant data are within the manuscript and its Supporting Information files.

## Abstract

### Background and purpose

Renal excretion of direct oral anticoagulants (DOACs) varies depending on the drug. Hypothetically, an increased glomerular filtration rate (GFR) may lead to suboptimal dosing and a higher thromboembolic events incidence. However, real-world patient data do not support the theoretical risk. The aim is to analyse DOAC outcomes in patients with normal and high (≥90 mL/min) GFR, focusing on biological parameters and thrombotic/haemorrhagic events.

### Methods

Observational prospective single-centre study and registry of patients on DOACs. Follow-up was 1,343 patient-years. A bivariate analysis was performed of baseline variables according to GFR (<90 mL/min vs ≥90 mL/min). Anti-Xa activity before and after drug intake (HemosIL, Liquid Anti-Xa, Werfen) was measured for edoxaban, apixaban, and rivaroxaban; diluted thrombin time for dabigatran (HEMOCLOT); and additionally, plasma concentrations in edoxaban (HemosIl, Liquid Anti-Xa suitably calibrated).

### Results

1,135 patients anticoagulated with DOACs were included and 152 patients with GFR ≥90 mL/min. Of 18 serious thrombotic complications during follow-up, 17 occurred in patients with GFR <90 mL/min, and 1 in a patient with GFR ≥90 mL/min. A higher incidence of complications was observed in patients with normal GFR, but the difference was not statistically significant (p>0.05). No statistically significant differences with clinical relevance were observed between the normal or supranormal groups in anti-Xa activity or in edoxaban plasma concentrations.

### Conclusions

There was no increased incidence of thrombotic/haemorrhagic complications in our patients treated with DOACs, including 66% treated with edoxaban, and patients with GFR ≥90 mL/

**Funding:** The HSCSP Haemostasis and Thrombosis and the IIB-Sant Pau receive funding from Daiichi-Sankyo to develop and maintain the MACACOD (Real-life Clinical Outcomes of Direct Oral Anticoagulants) registry, of which this study is part. Only the authors have participated in study design, data collection and analysis, and manuscript preparation. Dr Souto has received honoraria or financial support for travel, accommodation, or expenses from Laboratorios Rovi, Leo Pharma, Baxter, Sanofi, Boehringer Ingelheim, Pfizer, Bristol Myers Squibb, Roche, Daiichi-Sankyo, and Stago Laboratories. He also holds advisory position in Devicare. The remaining authors declare no conflicts of interest.

**Competing interests:** I have read the journal's policy and the authors of this manuscript have the following competing interests: "The HSCSP Haemostasis and Thrombosis and the IIB-Sant Pau receive funding from Daiichi-Sankyo to develop and maintain the MACACOD (Real-life Clinical Outcomes of Direct Oral Anticoagulants) registry, of which this study is part. Only the authors have participated in study design, data collection and analysis, and manuscript preparation. Dr Souto has received honoraria or financial support for travel, accommodation, or expenses from Laboratorios Rovi, Leo Pharma, Baxter, Sanofi, Boehringer Ingelheim, Pfizer, Bristol Myers Squibb, Roche, Daiichi-Sankyo, and Stago Laboratories. He also holds advisory position in Devicare. The remaining authors declare no conflicts of interest. This does not alter our adherence to PLOS ONE policies on sharing data and materials.

min. Likewise, drug anti-Xa activity and edoxaban plasma concentration did not seem to be influenced by GFR.

## Introduction

Since direct oral anticoagulant (DOAC) drugs are mainly eliminated by the kidneys, a patient's glomerular filtration rate (GFR) is an important dose reduction criterion. Edoxaban is 50% and 40% excreted unmetabolized by the kidney and liver, respectively, and the remaining 10% is metabolized by cytochrome CYP3A4/5 [1].

Regarding the impact of kidney function on DOAC outcomes, four pivotal clinical trials (RE-LY, with dabigatran; ROCKET AF, with rivaroxaban; ARISTOTLE, with apixaban; and ENGAGE AF-TIMI 48, with edoxaban) [2–5] compared efficacy and safety of different DOACs and warfarin. Patients were broadly grouped by creatinine clearance (CrCl) intervals into three main groups: CrCl <50 mL/min, CrCl 50–80 mL/min, and CrCl >80 mL/min (and while the outside limits differed somewhat between studies, the minimum was never below CrCl 30 mL/min). Results for DOAC efficacy and safety sub-analyses of the CrCl 30–80 mL/min group (30–95 mL/min for edoxaban) were consistent with the overall analyses, and with that of patients without chronic kidney disease (CKD). The same finding was reported for major outcomes, such as the risk of stroke or systemic embolism, major bleeding, intracranial haemorrhage (ICH), other bleeding, and mortality [6–9].

DOACs for patients with an increased GFR (CrCl >95 mL/min) have been reported to be less effective, as either the desired therapeutic range is not reached or the drug is more rapidly eliminated. However, increased GFR needs to be differentiated from hyperfiltration, observed fundamentally in critically ill acute patients (CrCl >120 mL/min), for whom decreased efficacy of other drugs, such as antibiotics, has been observed [10]; the same occurs in obese patients with diabetes, who typically have GFR $\geq$120 mL/min [11], although it is not clear whether this affects drug efficacy.

The ENGAGE AF-TIMI 48 study [5], which compared the efficacy and safety of edoxaban to warfarin in reducing the risk of stroke and systemic embolic events in patients with atrial fibrillation (AF), reported a higher ischaemic stroke rate for edoxaban in patients with CrCl >95 mL/min (hazard ratio (HR) [95% confidence interval (CI)]: 1.45 (0.90–2.35); p = 0.13). Consequently, the US Food and Drugs Administration (FDA) issued a warning not to use edoxaban for stroke prevention in patients with AF with CrCl >95 mL/min, recommending instead to use DOACs [12], while the European Medicines Agency (EMA) issued the recommendation that "edoxaban should be used only in patients with high CrCl after careful evaluation of individual thromboembolic and bleeding risk" [13].

In a subsequent analysis of ENGAGE AF-TIMI 48 data, Bohula et al [9] reported that the net clinical benefit of high-dose edoxaban and warfarin in preventing arterial thromboembolism were comparable across the entire kidney function range, based on the finding that, while relative efficacy and safety apparently decreased for the upper CrCl range (>95 mL/min: HR [95% CI]: 1.36 [0.88–2.10]), this decrease was not statistically significant (interaction p = 0.08).

Regarding other DOACs, a retrospective study of data from the ROCKET-AF clinical trial [8] interestingly reported that patients with CrCl >95 mL/min on rivaroxaban showed a higher (but not statistically significant) rate of stroke and systemic thromboembolism than patients on warfarin (HR [95% CI]: 1.47 [0.81–2.68]; p = 0.033). Kidney function analyses have also been performed for apixaban and dabigatran, but not for specific CrCl >95mL/min subgroups [6, 7]. For a first ischaemic stroke in patients with CrCl $\geq$80mL/min, an FDA

statistical analysis reported HR = 0.84 for dabigatran, HR = 1.07 for rivaroxaban, and HR = 1.35 for apixaban vs warfarin [14].

In a meta-analysis of the above-mentioned pivotal trials [2–5], DOACs compared to warfarin significantly reduced stroke and systemic embolic events (relative risk (RR) [95% CI]: 0·81 [0·73–0·91]; p<0·0001), with no subgroup differences observed for stroke or systemic embolic events. Note that, although not statistically significant, patients with CrCl ≥80 mL/min experienced more events compared to patients in the other CrCl ranges (<50 mL/min, RR [95% CI]: 0.79 [0.65–0.96]; 50–80 mL/min, RR [95% CI]: 0.75 [0.66–0.85]; >80 mL/min: RR [95% CI]: 0.98 [0.79–1.22]; interaction p = 0.12) [15].

Real-world evidence of the efficacy and safety profile of DOACs in routine clinical practice has generally been shown to be consistent with randomized clinical trials: compared to warfarin, all DOACs (including rivaroxaban, dabigatran, apixaban, and edoxaban) were associated with lower rates of ischaemic stroke, major bleeding, and mortality [16–21]. Nevertheless, few studies have analysed patients with high GFR (CrCl ≥95 mL/min), and what results have been reported for edoxaban are contradictory, as they describe both a higher and lower risk of ischaemic stroke and systemic embolism with low-dose edoxaban [20, 21].

The aim of this study was to compare, in real-life practice, characteristics, biological parameters (anti-Xa activity), and thrombotic/haemorrhagic complication rates in DOAC-treated patients with GFR ≥90 mL/min vs GFR <90mL/min.

## Materials and methods

An observational prospective single-centre study was conducted with patients treated at the Haemostasis and Thrombosis Unit of the Hospital de la Santa Creu i Sant Pau (HSCSP, Barcelona, Spain). The study, part of the Real-life Clinical Outcomes of Direct Oral Anticoagulants (MACACOD) project (NCT04042155), was conducted in accordance with the Declaration of Helsinki, the study protocol was approved by the HSCSP Ethics Committee, and included patients signed an informed consent.

Included were patients aged >18 years, with AF or venous thromboembolism (VTE), treated with DOACs to prevent stroke or systemic embolism due to their high VTE risk ($CHA_2DS_2$-VASc score ≥2) [22]. Patients were treated with DOACs following indications of the 2016 AEMPS Therapeutic Positioning Report UT_ACOD/V5/21112016 [23]. Note that DOAC use (and therefore our study inclusion and exclusion criteria) is determined by the fact that, in Spain, DOACs are only funded by the public health system in certain circumstances, while vitamin K antagonists (VKAs) continue to be the first therapeutic option.

### Inclusion criteria

- Patients with known hypersensitivity or specific contraindication to VKA use.

- Patients with a history of ICH before starting/during oral anticoagulant therapy with VKAs, in which case anticoagulation benefits were determined to outweigh bleeding risk.

- Patients who had experienced ischaemic stroke and were at high risk of ICH according to clinical and neuroimaging criteria (HAS-BLED score ≥3, grade III-IV leukoaraiosis, multiple cortical microbleeds).

- Patients in treatment with VKAs experiencing serious arterial thromboembolic events despite proper international normalized ratio (INR) control (time in the therapeutic range (TTR) >65%).

- Patients in treatment with VKAs for whom correct INR control was not possible despite good therapeutic compliance (TTR <65%).

- Patients for whom conventional INR control was not possible during the COVID-19 lock-down, which meant that they started treatment directly with DOACs.

## Exclusion criteria

- Patients in treatment with VKAs under good INR control.

- Except during COVID-19 peak pandemic moments, patients with newly diagnosed AF for whom anticoagulation was indicated (first time receiving oral anticoagulants).

- Patients with AF with severe cardiac valve involvement.

- Patients with significant cognitive impairment, psychiatric disorders, or alcohol-abuse disorders (unsupervised) whose collaboration could not be guaranteed.

- Patients for whom oral anticoagulants were contraindicated: pregnancy, severe acute bleeding in the previous month, recent (10 days prior), or planned central nervous system surgery, severe liver disease or CKD, CrCl <15 mL/min, severe or uncontrolled hypertension, and altered haemostasis (hereditary or acquired and with a significant bleeding risk).

## Recruitment and data collection

Patient recruitment began in July 2019 and data was collected up to March 2022. Procedures for visits, follow-up, and sample collection were the same as for non-included patients (i.e., usual clinical practice), as follows:

- First medical and specialized nurse visit. Baseline data collected included sex, age, height, weight, body mass index (BMI), Charlson comorbidity index (CCI) score, and also anticoagulation indication (VTE or AF), VTE risk for patients with AF ($CHA_2DS_2$-VASc), bleeding risk (HAS-BLED score), previous history of thrombosis/haemorrhage, possible contraindications, laboratory values, kidney and liver function values, and DOAC type and dosage. Note that drugs were prescribed independently of the study, and dosage was based on the corresponding technical data sheet. At this visit, patients signed their informed consent to participate in the study.

- Educational session. The patients received individual or group training on anticoagulants, covering topics such as the importance of adherence, how to respond to complications, information on invasive procedures, drug interactions, etc.

- Follow-up. Face-to-face visits were scheduled for approximately four weeks after starting DOAC treatment. Patients with progressive renal failure, at high VTE risk, and at intermediate-low risk VTE risk were also scheduled for quarterly, twice yearly, and annual visits, respectively. Recorded for each visit were laboratory data (basic coagulation study, blood count, kidney function, and liver function), thrombotic complications since the previous visit (stroke or systemic embolism: type, date, location, and severity), and haemorrhagic complications scored according to the BARC scale [24].

## Samples

In the first follow-up visit, pre- (trough) and post- (two hours after peak) drug intake anti-Xa activity (HemosIL, Liquid Anti-Xa, Werfen) was determined for patients on edoxaban,

apixaban, and rivaroxaban; diluted thrombin time (dTT) was measured (HEMOCLOT Thrombin Inhibitors, Hyphen BioMed, Neuville-sur-Oise, France) for patients on dabigatran; and plasma concentrations (HemosIL, Liquid Anti-Xa suitably calibrated) were measured for patients on edoxaban.

## Statistical analysis

Patients were divided into GFR groups (using the Cockcroft-Gault formula) according to their CrCl values: <90 vs ≥90 mL/min, and ≤30 vs >30–50 vs >50–90 vs >90–110 vs >110–130 and >130 mL/min.

Bivariate analysis was used to compare the different baseline variables for the groups. Categorical variables were compared using the Chi-square or Fisher's exact test, as appropriate, and continuous variables were compared using the Mann-Whitney U test. Thromboembolic/ haemorrhagic complication groups were analysed using Fisher's exact test. In all cases, values of $p < 0.05$ were considered statistically significant.

In the subgroup of patients treated with edoxaban, associations between pre- and post-dose anti-Xa activity and plasma drug concentrations were analysed using Spearman's rho correlation coefficient, and the relationships between anti-Xa activity and age, BMI, GFR, and drug dose were analysed using Fisher's exact test or Spearman's rho, depending on the nature of the variable (categorical or continuous).

All analyses were performed using SPSS 26.0.

## Results

A total of 1,135 patients were included in the study, 1,098 receiving anticoagulation for AF and 37 for VTE; 121 (10.7%) were on dabigatran, 230 (20.3%) on apixaban, 749 (66.0%) on edoxaban, and 35 (3.1%) on rivaroxaban. Table 1 shows the baseline characteristics of the patients by kidney function group. The cutoff point was set at 90 mL/min, on the basis that GFR ≥90 mL/min reflected normal or increased GFR, and GFR ≤90 mL/min reflected some degree of kidney function impairment.

Both groups had similar CCI scores (p = 0.438), and similar post-dose (p = 0.801) anti-Xa activity for rivaroxaban, apixaban, and edoxaban. Post-dose anti-IIa activity levels for dabigatran were also similar. Despite having significant differences in anti-Xa pre-dose (p = 0.006) and anti-IIa post-dose (p = 0.050), it doesn't have any important clinical relevance. As for differences in terms of sex, age, and BMI (p<0.001), the GFR <90 mL/min group had more women, was 12 years older on average, had slightly lower BMI values, and had a $CHA_2DS_2$-VASc score on average one point higher, indicating a higher VTE risk. No significant differences were observed regarding pre-dose (p = 0.178) and post-dose (p = 0.312) plasma concentrations of edoxaban, measured in 407 patients. Table 2 shows renal function distributions according to each drug.

### Thrombotic/haemorrhagic complications

Follow-up in patient-years was 1,342.71 (median [P25-P75] 12.8 [7.6–21.2] months). Serious thrombotic/haemorrhagic complications were analysed for 1,135 patients who had at least one follow-up visit in addition to the baseline visit. Table 3 summarizes details of thrombotic/haemorrhagic complications.

Of 18 serious thrombotic complications (10 with edoxaban, 6 with apixaban, and 2 with dabigatran), 17 occurred in the GFR <90 mL/min group. In this group, follow-up was 1,196.95 patient-years (median 13.2 months) and incidence was 1.42 events/100 patient-years (95% CI: 0.83–2.27). In the GFR ≥90 mL/min group, follow-up was 145.76 patient-years

**Table 1. Patient baseline characteristics by GFR.**

| | GFR <90 mL/min (n = 983) | GFR ≥90 mL/min (n = 152) | p |
|---|---|---|---|
| **Follow-up in years,** total | 1196.95 | 145.76 | |
| **Follow-up in months,** median [P25—P75] | 13.2 [8.0–21.8] | 9.7 [6.5–15.1] | <0.001 |
| **Sex** (men), % | 488 (49.6%) | 118 (77.6%) | <0.001 |
| **Age** (years), mean (SD) | 79.2 (7.5) | 67.1 (10.2) | <0.001 |
| **CCI,** mean (SD) | 1.53 (1.45) | 1.44 (1.42) | 0.438 |
| **BMI,** mean (SD) | 26.4 (4.4) | 31.7 (5.6) | <0.001 |
| **CHA$_2$DS$_2$-VASc,** mean (SD) | 4.19 (1.53) | 2.87 (1.68) | <0.001 |
| **Pre-dose anti-Xa** (UI/mL), median [P25—P75] | 0.12 [0.06–0.21] * | 0.09 [0.04–0.17] * | 0.006 |
| **Post-dose anti-Xa** (UI/mL), mean [P25—P75] | 1.28 [0.93–1.73] * | 1.30 [1.02–1.62] * | 0.801 |
| **Patients on dabigatran** | **n = 19** | **n = 6** | |
| **Pre-dose anti-IIa** (ng/mL), median [p25-p75] | 100 [90–100] | 110 [100–130] | 0.176 |
| **Post-dose anti-IIa** (ng/mL), median [p25-p75] | 160 [100–300] | 300 [205–410] | 0.050 |
| **Edoxaban plasma concentrations** | **n = 363** | **n = 44** | |
| **Pre-dose** (ng/mL), mean (SD) | 33.7 (25.0) | 31.2 (24.2) | 0.178 |
| **Post-dose** (ng/mL), mean (SD) | 239.8 (89.1) | 254.3 (92.0) | 0.312 |

*In the GFR <90mL/min group, pre-dose and post-dose anti-Xa activity was measured in 822 patients. In the GFR ≥90mL/min group, pre-dose and post-dose anti-Xa activity was measured in 110 patients.

**Table 2. GFR by drug.**

| GFR mL/min | Edoxaban | | Apixaban | | Dabigatran | | Rivaroxaban | |
|---|---|---|---|---|---|---|---|---|
| | n | (%) | n | (%) | n | (%) | n | (%) |
| <30 | 18 | (2.4%) | 26 | (11.3%) | 3 | (2.5%) | 1 | (2.9%) |
| 30–50 | 213 | (28.4%) | 77 | (33.5%) | 22 | (18.2%) | 8 | (22.9%) |
| 50–90 | 425 | (56.7%) | 99 | (43.1%) | 71 | (58.7%) | 20 | (57.2%) |
| 90–110 | 62 | (8.3%) | 13 | (5.7%) | 13 | (10.7%) | 3 | (8.6%) |
| 110–130 | 21 | (2.8%) | 8 | (3.5%) | 9 | (7.4%) | 0 | (0%) |
| >130 | 10 | (1.3%) | 7 | (3.0%) | 3 | (2.5%) | 3 | (8.6%) |
| **Total** | 749 | (100%) | 230 | (100%) | 121 | (100%) | 35 | (100%) |

**Table 3. Complications by GFR: Number/incidence (% patient-year).**

| | GFR <90mL/min(n = 898) | | GFR ≥90mL/min(n = 132) | | p |
|---|---|---|---|---|---|
| | N | Incidence (95% CI) | N | Incidence (95% CI) | |
| **Thromboembolic** | 17 | 1.42 (0.83–2.27) | 1 | 0.69 (0.02–3.82) | 0.470 |
| **Major haemorrhagic** | 38 | 3.18 (2.25–4.36) | 5 | 3.43 (1.11–8.01) | 0.871 |
| **Total major complications** | 55 | 4.60 (3.46–5.98) | 6 | 4.12 (1.51–8.96) | 0.798 |
| **CRNMB** | 97 | 8.10 (6.57–9.89) | 9 | 6.18 (2.82–11.72) | 0.434 |
| **Mortality** | 76 | 6.35 (5.00–7.95) | 6 | 4.12 (1.51–8.96) | 0.303 |

(median 9.7 months) and incidence was 0.69 events/100 patient-years (95% CI: 0.02–3.82). Incidences between both groups were not significantly different (p = 0.470).

There were 149 haemorrhagic complications, 109 with edoxaban, 27 with apixaban, 12 with dabigatran and 1 with rivaroxaban. Most bleeds (n = 106) were classified as clinically relevant non-major bleeding (CRNMB), i.e., BARC grade 2; 43 were BARC grade 3 or more (severe),

and 1 bleed resulted in death. Of the 43 severe bleeds, 38 occurred in the GFR <90 mL/min group, and 5 in the GFR ≥90 mL/min group. In the GFR <90 mL/min group, incidence was 3.18 events/100 patient-years (95% CI: 2.25–4.36) vs 3.43 events/100 patient-years (95% CI: 1.11–8.01) in the GFR ≥90 mL/min group. Incidences between both groups were not significantly different (p = 0.871).

No statistically significant differences were found between the groups on analysing the relationship between renal function and total major (thromboembolic plus haemorrhagic) complications (p = 0.798). Most events occurred in the larger GFR <90 mL/min group, while the numbers of events was very low in the other group. Incidence rates therefore need to be interpreted with care.

## Discussion

The main hypothesis on which the FDA [11] and EMA [12] based their recommendations regarding edoxaban was that fixed doses in patients with supranormal renal function (CrCl >95 mL/min) may be less efficacious in preventing VTE events. There is no such recommendation regarding other DOACs, although a similar effect has been shown with rivaroxaban [8], while no efficacy or safety sub-analyses of dabigatran or apixaban have been conducted for this population.

Real-world data regarding high GFR and its outcomes in patients on DOACs is scarce. Yu et al [20] reported that, at CrCl >95 mL/min, low-dose edoxaban was less effective in preventing ischaemic stroke and systemic embolism compared with warfarin (interaction p = 0.023); noteworthy was the fact that the number of complications was few (2 for edoxaban and 2 for warfarin), for HR = 1.41 and a very wide 95% CI (0.16–8.10). On the other hand, Lee et al [21] found that the incidence of ischaemic stroke with edoxaban in patients with high, while the incidence of normal renal function (CrCl >95 mL/min) was lower than that for warfarin (2.20/100 person-years vs 3.04/100 person-years), although the difference was not statistically significant; in that study, 56% of patients on edoxaban were prescribed the 30 mg dose. In a recent comparison of edoxaban 60 mg and edoxaban 75 mg once daily in patients with CrCl >100 mL/min, Yin et al. [25] reported a similar risk of overall stroke and major/clinically relevant bleeding for both treatments, concluding that edoxaban 60mg/24h was effective and safe for this patient profile.

In our study, as in most real-life scenarios, we observed that once we grouped the population by renal function, baseline characteristics were not homogeneous. As expected, given the epidemiology of CKD, patients with normal or supranormal renal function were mostly men and younger than patients with impaired kidney function.

Although edoxaban is not approved for patients with CrCl >95 mL/min in the USA [16], our data suggest that there are no significant differences in complication rates between other DOACs and edoxaban for the GFR <90 mL/min group compared to the GFR ≥90mL/min group. Moreover, unlike other studies, our study reports anti-Xa activity measurement for both those groups.

We found that the overall incidence of serious thrombotic/haemorrhagic events (mainly gastrointestinal) for all DOACs was very low compared to findings for warfarin [26, 27]. We found no significant difference in the incidence of thrombotic complications between patients with normal/supranormal renal function and those with impaired renal function, nor did we find significant differences for post-dose anti-Xa activity, pre-dose anti-IIa activity, and plasma concentrations of edoxaban for patients with GFR <90 mL/min compared to patients with GFR ≥90 mL/min. Regarding pre-dose anti-Xa and post-dose anti-IIa the significant differences (p = 0.006 and p = 0.050 respectively) were not important when translated into clinical

results. Also we have to take into account that in the anti-IIa measurements the number of patients is really small (n = 25).

Note that both the anti-Xa and drug-calibrated dTT assays are considered to be comparable to mass spectrometry measurement of DOACs [28]; this is important because the assays are available in most hospital laboratories, and moreover, can be used for possible overdoses, before surgery, for patients with obesity or intestinal malabsorption, or to initiate thrombolytic post-stroke therapy [29].

The low number of complications in our population, along with the fact that biological parameters (anti-Xa activity, dTT, and plasma concentrations) were comparable, independently of the GFR, leads us to believe that DOAC plasma concentrations (if dosage is correct) are not associated with kidney function measured in CrCl terms, as reported elsewhere [30, 31].

The main strengths of our study are that it was prospective and was conducted in a single centre as part of routine clinical practice, thereby ensuring that procedures (visits, laboratory testing, and follow-up) were homogenous. The fact that the study was conducted at a single centre also ensured even protocol application to all patients, thanks to nursing-staff training provided to patients and their families. In addition, face-to-face follow-up with special emphasis on complications was ensured to avoid underestimating the main outcomes, while measurement of the anticoagulant effect of each DOAC at trough and peak points ensured objective comparison across kidney function subgroups by CrCl strata. The main limitation of this study is its relatively small sample size compared to large clinical trials, and a relatively short follow-up period. However, this is also strength, as this made personalized follow-up of patients possible and so was a good guarantee of complications being reported.

In conclusion, in our experience, GFR is not associated with clinical outcomes in patients treated with DOACs. Patients with a high or very high GFR experienced no increase in severe complications (specifically thromboembolic), not even on edoxaban. Conversely, we found no differences in drug plasmatic effects between patients with normal/supranormal and impaired kidney function. We report a satisfactory response in our population to dose adjustment, as recommended for moderately and severely impaired renal function (GFR <50 mL/min), with plasma drug levels resulting homogeneous for the different GFR values.

## Supporting information

**S1 Data.**
(XLS)

**S2 Data.**
(XLS)

**S3 Data.**
(XLS)

## Author Contributions

**Conceptualization:** René Acosta-Isaac, Sergi Mojal, Carla Moret, Joan Carles Souto.

**Data curation:** René Acosta-Isaac, Sergi Mojal.

**Formal analysis:** Sergi Mojal, Joan Carles Souto.

**Funding acquisition:** Joan Carles Souto.

**Investigation:** Mariana Corrochano, René Acosta-Isaac, Melania Plaza, Sergi Mojal, Carla Moret, Joan Carles Souto.

**Methodology:** Mariana Corrochano, René Acosta-Isaac, Melania Plaza, Rodrigo Muñoz, Sergi Mojal, Carla Moret, Joan Carles Souto.

**Project administration:** Mariana Corrochano, René Acosta-Isaac, Carla Moret, Joan Carles Souto.

**Resources:** René Acosta-Isaac, Joan Carles Souto.

**Software:** Sergi Mojal.

**Supervision:** Mariana Corrochano, René Acosta-Isaac, Melania Plaza, Carla Moret, Joan Carles Souto.

**Validation:** René Acosta-Isaac, Melania Plaza, Sergi Mojal, Carla Moret, Joan Carles Souto.

**Visualization:** Mariana Corrochano, Melania Plaza, Rodrigo Muñoz, Carla Moret, Joan Carles Souto.

**Writing – original draft:** Mariana Corrochano, Melania Plaza, Rodrigo Muñoz.

**Writing – review & editing:** Mariana Corrochano, René Acosta-Isaac, Melania Plaza, Rodrigo Muñoz, Sergi Mojal, Carla Moret, Joan Carles Souto.

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
