## [Decision Letter · Decision Letter 0]

28 Aug 2022

PONE-D-22-15972Impact of increased kidney function on clinical and biological outcomes in real-world patients treated with Direct Oral AnticoagulantsPLOS ONE

Dear Dr. Plaza,

Thank you for submitting your manuscript to PLOS ONE. After careful consideration, we feel that it has merit but does not fully meet PLOS ONE’s publication criteria as it currently stands. Therefore, we invite you to submit a revised version of the manuscript that addresses the points raised during the review process.

We look forward to receiving your revised manuscript.

Kind regards,

Sreeram V. Ramagopalan

Academic Editor

PLOS ONE

https://journals.plos.org/plosone/s/file?id=ba62/PLOSOne_formatting_sample_title_authors_affiliations.pdf".

“This study was funded by an unconditional grant from Daiichi-Sankyo Spain.”

“The HSCSP Haemostasis and Thrombosis and the IIB-Sant Pau receive funding from Daiichi-Sankyo to develop and maintain the MACACOD (Real-life Clinical Outcomes of Direct Oral Anticoagulants) registry, of which this study is part. Only the authors have participated in study design, data collection and analysis, and manuscript preparation. Dr Souto has received honoraria or financial support for travel, accommodation, or expenses from Laboratorios Rovi, Leo Pharma, Baxter, Sanofi, Boehringer Ingelheim, Pfizer, Bristol Myers Squibb, Roche, Daiichi-Sankyo, and Stago Laboratories. He also holds advisory position in Devicare. The remaining authors declare no conflicts of interest.”

“The HSCSP Haemostasis and Thrombosis and the IIB-Sant Pau receive funding from Daiichi-Sankyo to develop and maintain the MACACOD (Real-life Clinical Outcomes of Direct Oral Anticoagulants) registry, of which this study is part. Only the authors have participated in study design, data collection and analysis, and manuscript preparation. Dr Souto has received honoraria or financial support for travel, accommodation, or expenses from Laboratorios Rovi, Leo Pharma, Baxter, Sanofi, Boehringer Ingelheim, Pfizer, Bristol Myers Squibb, Roche, Daiichi-Sankyo, and Stago Laboratories. He also holds advisory position in Devicare. The remaining authors declare no conflicts of interest.”

Please confirm that this does not alter your adherence to all PLOS ONE policies on sharing data and materials, by including the following statement: ""This does not alter our adherence to  PLOS ONE policies on sharing data and materials.” (as detailed online in our guide for authors http://journals.plos.org/plosone/s/competing-interests).  If there are restrictions on sharing of data

and/or materials, please state these. Please note that we cannot proceed with consideration of your article until this information has been declared.

Reviewers' comments:

Reviewer's Responses to Questions

**Comments to the Author**

1. Is the manuscript technically sound, and do the data support the conclusions?

Reviewer #1: Partly

2. Has the statistical analysis been performed appropriately and rigorously? 

Reviewer #1: No

3. Have the authors made all data underlying the findings in their manuscript fully available?

Reviewer #1: No

4. Is the manuscript presented in an intelligible fashion and written in standard English?

Reviewer #1: No

5. Review Comments to the Author

Reviewer #1: Corrochano et al compared occurrence of several clinical events and DOAC concentrations (or anti-Xa activity) between patients treated with DOAC with different ranges of GFR. They specifically focused on whether the patients with supranormal kidney function (i.e., GRP ≥90 mL/min) would experience different prognosis and DOAC concentrations (compared to those with “normal” kidney function). Below are my comments on the manuscript:

1. The authors should notice the difference between (1) whether the superiority (or at least non-inferiority) of DOAC versus warfarin differs between patients with supranormal and normal kidney function levels and (2) whether efficacy of (either overall, or a specific type of) DOAC differs between patients with supranormal and normal kidney function levels. What the authors examined is actually the latter one, which is more relevant to investigate but obviously requires a larger sample size. The study is thus at risk of being underpowered. An analysis of statistical power is suggested as it is very likely that there could be difference of clinical outcomes between supranormal and normal kidney function levels, but the study failed to detect this simply due to the limited sample size. This limitation should also be discussed more in the discussion section, and the conclusion should be modified, because absence of evidence is not evidence of absence. For example, the authors mentioned “Although edoxaban is not approved for patients with CrCl >95 mL/min in the USA [16], our data suggest that there are no significant differences in complication rates …”, but I don’t think the current study can serve as evidence against the FDA recommendation.

2. According to the inclusion/exclusion criteria presented in the method section, the study population is highly selected. This should be mentioned as a potential source of selection bias (i.e., limited generalizability). The study population mainly consisted of patients who were not “well” treated by VKA, while based on current international guidelines, DOAC is often the first recommended oral anticoagulant.

3. The study population were actually with GFR ≥30 mL/min, while in the manuscript it is mentioned several times that the investigated patients were with normal or supranormal kidney functions. Could a GFR <90 mL/min be termed as normal? This is misleading and should be corrected. It also raises a concern that it might be not appropriate to compare patients with GFR ≥90 mL/min to those with decreased GFR (given the research question the authors aimed to answer).

4. More details should be added into the method section. The most important one is about the follow-up. The study seems a cohort study, and I assume the authors evaluated the many inclusion/exclusion criteria at baseline. If so, would the patients be evaluated during the follow-up regarding the inclusion/exclusion criteria? When would a patient be censored? In real practice, patients may switch from DOAC to VKA, or stop receiving DOAC due to some medical reasons, and GFR may change as well. How did the authors handle these situations? The patients with GFR ≥90 mL/min had much shorter follow-up than those with GFR <90 mL/min, which is very likely to introduce bias (e.g., fewer clinical events were identified simply due to the shorter follow-up). In addition, more details should be introduced about how the studied clinical outcomes were determined (i.e., diagnosis criteria, recurrence, etc).

5. The comparison of clinical outcomes between the two groups is at great risk of confounding. The statistical method the authors used is not helpful for this issue.

6. Minor points:

- Title: I suggest the authors consider to use a title which is less causal, such as “Drug levels and clinical outcomes of patients treated with direct oral anticoagulants with supranormal kidney function”.

- Keywords: Why did the authors only present edoxaban as a keyword but not the other three investigated DOACs?

- Manuscript: About the presenting sequence of DOAC concentrations (or anti-Xa activity) and clinical outcomes, intuitively it is better to present DOAC concentrations first and then clinical outcomes, as the purpose of the abstract described “… focusing on biological parameters and thrombotic/haemorrhagic events”. At least, keep the sequence consistent throughout the manuscript.

- Abstract: It is more informative to report the median follow-up, and report it in the result section instead of the method section.

- Abstract: Outcome events that were investigated were not described (i.e., defined) in the method section. This makes it confusing in the result section that “A higher incidence of complications was observed…” In addition, haemorrhagic complications were only mentioned in the conclusion.

- Abstract: The “normal” range of GFR should be presented, instead of “<90mL/min” only.

- Abstract: The percentages of serious thrombotic complications should also be presented for both groups, instead of the absolute numbers only.

- Abstract: It is still informative to present some numbers of the results of DOAC concentrations (or anti-Xa activity) between groups in the abstract, instead of simply mentioning “No statistically significant differences”, as the study is at risk of being underpowered.

- Abstract: Please rephrase the conclusion in the abstract. Some details are better to present in the result section, such as “including 66% treated with edoxaban”.

- Introduction: How other types of DOACs are eliminated should also be introduced.

- Introduction: The reference 5 was not the ENGAGE AF-TIMI 48 trial. “The ENGAGE AF-TIMI 48 study [5], …, reported a higher ischaemic stroke rate for edoxaban in patients with CrCl >95 mL/min (hazard ratio (HR) [95% confidence interval (CI)]: 1.45 (0.90-2.35); p=0.13).” This sentence and the HR here is difficult to understand without reading the manuscript that reported the trial, as the trial compared Edoxaban with Warfarin, while the interested comparison for the current manuscript is Edoxaban in patients with supranormal GFR versus normal GFR who are both treated with Edoxaban (see my first comment). For the same reason, the paragraph 2, 5, and 6 of the introduction are not that relevant to the current study. In addition, I suggest to introduce more evidence based on which FDA and EMA made the recommendations, and explain why the current study is still relevant to conduct given the existing recommendations.

- Introduction: “DOACs for patients with an increased GFR (CrCl >95 mL/min) have been reported to be less effective, as either the desired therapeutic range is not reached or the drug is more rapidly eliminated.” Citations of these reports are necessary.

- Method: Please add the ethic approval number for the study. If it is the same as that of the project number, this should be mentioned in the method section.

- Method: It is incorrect that the CHA2DS2-VASc score is used for evaluating risk of VTE (instead, for ischemic stroke in AF patients). The below text in the manuscript should be corrected: “… treated with DOACs to prevent stroke or systemic embolism due to their high VTE risk (CHA2DS2-VASc score ≥2)”; “… VTE risk for patients with AF (CHA2DS2-VASc)…”; “… and had a CHA2DS2-VASc score on average one point higher, indicating a higher VTE risk.”

- Discussion: I don’t think it’s correct that “(limited sample size and short follow-up) is also strength, as this made personalized follow-up of patients possible and so was a good guarantee of complications being reported”.

- Manuscript: There are some grammar errors, or sentences that seem not natural to the English language. I list some of them below, but please carefully read the full text and try to improve the language.

--- Abstract: “… a higher thromboembolic events incidence” -> ““… a higher incidence of thromboembolic events”;

--- Abstract: “Observational prospective single-centre study and registry of patients on DOACs.” This is not a sentence;

--- Abstract: “Anti-Xa activity … was measured for edoxaban, apixaban, and rivaroxaban; diluted thrombin time for dabigatran …; and additionally, plasma concentrations in edoxaban …” Please rephrase this sentence;

--- Introduction: “Since direct oral anticoagulant (DOAC) drugs” -> remove the “drugs”;

--- Method: “… and included patients signed an informed consent” -> “… and included patients who signed an informed consent”.

6. PLOS authors have the option to publish the peer review history of their article (what does this mean?). If published, this will include your full peer review and any attached files.

Reviewer #1: No

---

## [Author Response · Author response to Decision Letter 0]

4 Nov 2022

All the responses to each editor and reviewer comment can be found in the file called "Response to Reviewers".

---

## [Decision Letter · Decision Letter 1]

22 Nov 2022

Drug levels and clinical outcomes of patients treated with direct oral anticoagulants with supranormal kidney function

PONE-D-22-15972R1

Dear Dr. Plaza,

We’re pleased to inform you that your manuscript has been judged scientifically suitable for publication and will be formally accepted for publication once it meets all outstanding technical requirements.

Kind regards,

Sreeram V. Ramagopalan

Academic Editor

PLOS ONE

Additional Editor Comments (optional):

Reviewers' comments:

Reviewer's Responses to Questions

**Comments to the Author**

1. If the authors have adequately addressed your comments raised in a previous round of review and you feel that this manuscript is now acceptable for publication, you may indicate that here to bypass the “Comments to the Author” section, enter your conflict of interest statement in the “Confidential to Editor” section, and submit your "Accept" recommendation.

Reviewer #1: All comments have been addressed

2. Is the manuscript technically sound, and do the data support the conclusions?

Reviewer #1: Yes

3. Has the statistical analysis been performed appropriately and rigorously? 

Reviewer #1: Yes

4. Have the authors made all data underlying the findings in their manuscript fully available?

Reviewer #1: Yes

5. Is the manuscript presented in an intelligible fashion and written in standard English?

Reviewer #1: Yes

6. Review Comments to the Author

Reviewer #1: I appreciate the authors’ revisions of the manuscript after taking my comments into account, and I think the manuscript now meets the criteria of publication.

Below are some extra minor revisions that can be considered (based on the PDF version with track change), but an extra review is not necessary.

1. It’s better to mention “supranormal kidney function” in the short title. For example, “DOAC use in patients with supranormal kidney function”.

2. It is mentioned “¶ All the authors contributed equally to this work.” in the title page. However, it is unclear which authors the statement refers to.

3. Abstract: “An observational prospective single-center study and registry of patients on DOACs was performed.” -> “An observational prospective single-center study using registry of patients on DOACs was performed.”

4. Abstract: “All outcomes were diagnosed in the hospital by standard imaging methods.” -> “All outcomes (i.e., thromboembolic and hemorrhagic complications) were diagnosed in the hospital by standard imaging methods.”

5. Introduction: line 101 (page 36 of the PDF file): The FDA mentioned that “… This sentence missed the second quotation mark.

6. Methods: “Patients who switched from DOAC to VKA or simply stopped receiving DOAC and continued without anticoagulation ended the follow-up.” -> “The follow-up ended when patients switched from DOAC to VKA or simply stopped receiving DOAC and continued without anticoagulation.”

7. PLOS authors have the option to publish the peer review history of their article (what does this mean?). If published, this will include your full peer review and any attached files.

Reviewer #1: No

---

## [Editor Report · Acceptance letter]

1 Dec 2022

PONE-D-22-15972R1 

Impact of increased kidney function on clinical and biological outcomes in real-world patients treated with Direct Oral Anticoagulants 

Dear Dr. Plaza:

I'm pleased to inform you that your manuscript has been deemed suitable for publication in PLOS ONE. Congratulations! Your manuscript is now with our production department. 

Kind regards, 

on behalf of

Dr. Sreeram V. Ramagopalan 

Academic Editor

PLOS ONE